# Study on the Bandgap Characteristics and Vibration-Reduction Mechanism of Symmetric Power-Exponent Prismatic Phononic Crystal Plates

**Xing Jin and Zhenhua Zhang \***

College of Naval Architecture and Ocean, Naval University of Engineering, Wuhan 430033, China
\* Correspondence: 2zsz@163.com

**Abstract:** In this paper, a symmetric power-exponent prismatic phononic crystal configuration was proposed for the vibration reduction of thin plate structures, and the mechanism of bandgap generation and the influencing factors of the band gaps were analyzed. The results showed that the proposed symmetric power-exponent prismatic phononic crystal structure has three complete band gaps of bending waves, where the width of the second band gap can go up to 1639 Hz. The band gaps of bending waves of the phononic crystal were verified using a combination of numerical simulations and experimental methods, and subsequently, the bandgap characteristics and energy-focusing effect of the phononic crystals were effectively used to suppress the bending vibration of the thin plate. With the increase in prismoid height of the structure, the width of the first band gap expanded, while the bandwidths of the other two band gaps narrowed down. It was observed that an increase in the power of the power-exponent prismoid would reduce the starting and ending frequencies of the band gaps, whereas an increase in the prismoid edge thickness would weaken the energy-focusing effect and narrow the band gaps gradually. Our research results provide a new technique and a pathway to realize vibration reduction in thin plate structures.

**Keywords:** thin plates; phononic crystals; complete band gap; bending waves

## 1. Introduction

Plate and shell structures are widely used in many fields such as ships, aviation, and vehicles, and their vibration and noise reduction problems have always been an important concern in practical engineering applications. In this regard, the concept of acoustic metamaterials provides a new technological approach for the vibration and noise reduction of thin plate structures. An acoustic blackhole (ABH) can reduce the thickness of the structure by following a power exponential function ($h(x) = \varepsilon |x|^m (m \geq 2)$) [1], thus making the vibration wave energy concentrated at the tip of the structure. By virtue of additional damping materials, the vibration energy can be absorbed, thus achieving the vibration reduction. Therefore, ABH has a broad application prospect in the vibration reduction [2–4] and energy regulation [5–7] of beams or plates.

In recent years, a great deal of research has been carried out on ABH structures. Zhu et al. used the plane wave expansion method and finite element method to study an ABH-embedded thin plate structure, and found that the ABH could produce a birefringent effect on the bending waves [8]. In another work, Zhao et al. proposed an improved ABH plate structure, and verified through numerical simulation and emulation that ABH could produce collimating and focusing effects on the bending waves [9]. Zhou et al. studied the composite plate structure embedded with ABH via semi-analytical analysis, and the results showed that the ABH plate structure could effectively absorb the wave energy in the low-frequency band range with a relatively wide frequency band [10]. Du et al. studied the plate structure embedded with an acoustic blackhole through experiments and simulations, and the results revealed that the ABH plate attached with the damping

materials could effectively reduce the amplitude of sound pressure in the frequency band of 3600–5000 Hz [11]. Progressively, Tang et al. used the wavelet decomposition energy method to study a Euler–Bernoulli beam embedded with multiple ABHs, and observed that the ABH effect resulted in the generation of low-frequency band gaps of the bending waves [12]. Gao et al. proposed a beam structure with composite ABHs embedded in the beam center. They observed two band gaps at the frequency of 1200 Hz, and pointed out that the first band gap was generated by the coupling effect of longitudinal and transverse bending vibrations [13]. Deng et al. proposed a cylindrical shell structure embedded within ABHs and added stiffeners to enhance the shell stiffness. Their results showed that the proposed structure could effectively obstruct the axial propagation of bending waves in the shell [14]. In addition, they also used Gaussian expansion method to study the plate structure embedded with cross ABH grooves, and suppressed the bending vibration of the plates by utilizing the band gap characteristics of the proposed structure [15]. Tang et al. proposed a beam structure with embedded bi-bladed ABHs and produced a wide band gap by a combination of a local resonance effect and Bragg scattering effect [16]. Zhou et al. studied the dynamic and static characteristics of a composite ABH beam through numerical simulations, where the results showed that the composite ABH beam was superior to the traditional acoustic blackhole beam structure in terms of structural strength and vibration suppression [17]. O'Boy et al. used the Rayleigh–Ritz variational energy method to study the natural frequency of an ABH plate, and found that adding appropriate damping to the grooves could provide a better vibration-reduction effect than attaching the damping to the whole plate [18]. Ji et al. proposed a circular ABH absorber structure that could be attached to the plate structure, and verified its good vibration-reduction effect through the finite element method and experiments [19].

Phononic crystals also have a good application value in vibration and noise reduction, owing to their unique bandgap characteristics. Liu et al. proposed a phononic crystal of a lead ball covered by viscoelastic materials, and found that the wavelength corresponding to the phononic crystal band gap was much larger than the lattice size, essentially breaking through the limitation of the Bragg scattering mechanism and proposing a local resonance mechanism [20]. Ma et al. proposed a ring-like phononic crystal structure with a viscoelastic damping layer, verified the bandgap characteristics of such a structure by experiments and numerical simulations, and finally determined the relationship between the structural parameters and the band gap using the response surface method [21]. An et al. studied a two-dimensional cylindrical phononic crystal structure with periodic structures in radial and circumferential directions, and inhibited the propagation of radial waves in the structure by utilizing its bandgap characteristics [22]. Mourad et al. proposed a locally resonant plate structure with periodic silicone rubber arrays, and found that the structure possessed a complete band gap in the 1.9–2.6 kHz frequency band, where the band gap could be regulated by changing the height of the silicone rubber [23]. Ruan et al. proposed a spiral locally resonant phononic crystal structure and found that the structure could effectively suppress the bending vibration of the plate in the low-frequency band of 15–45 Hz [24]. Xiao et al. designed a cantilever-beam-type locally resonant phononic crystal, calculated its dispersion curve using the plane wave expansion method, and verified the bandgap characteristics of the proposed phononic crystal through experiments [25]. Sun et al. studied phononic crystal supercell structures with different filling rates, and the results showed that the supercell structure composed of protocells with different filling rates had band gaps in a lower frequency band compared to a simple protocell structure [26]. Moreover, Han et al. realized the directional transmission effect of sound waves by using bandgap characteristics and line defects of the phononic crystals [27].

The basic principles of ABH are to change the phase velocity and group velocity of the bending wave propagating in the structure, with the help of the changes in internal impedance of the structure. Thus, the wave energy can be concentrated in local areas of the structure, and then dissipated through a small amount of energy dissipating materials. Although many studies on ABHs have been reported, there exist only a few studies on

using bandgap characteristics of periodic ABHs to achieve the structure vibration reduction. In addition, most of the proposed configurations only have directional band gaps [28], while the literature on suppressing the bending vibration of the plate structures by using the periodic arrangement of power exponential structures to form complete band gaps is scarce. The traditional ABH configuration usually reduces the structure thickness in the form of a power exponential function, which degrades the structure strength. This paper proposes a symmetric power-exponent prismatic phononic crystal structure based on the ABH configuration. The power-exponent prismatic structure made of lightweight materials could be attached to thin plates by means of bonding, and the bending vibration of the plate structure could be effectively suppressed by using the energy focusing effect and bandgap characteristics of the phononic crystal structure. It was found that the phononic crystal had three complete wideband band gaps of bending waves. Based on the modal shapes and dispersion curves, the mechanism of the band gap generation was analyzed, and the influencing factors of the band gaps were studied. Finally, the bandgap characteristics of the structure were verified by numerical simulations and experiments.

## 2. Research Model

As shown in Figure 1, the proposed protocell of phononic crystal consists of two identical power-exponent prismoids and a thin plate, where the prismoids are attached to the upper and lower sides of the thin plate. The upper and lower end faces of the power-exponent prismoids are square in shape, and the prismoid faces bend inward, reflecting a power exponential curve. For the prismoid on the upper-end face, a cylindrical hole is set in the center of the face, as shown in Figure 2. The square side length of the power-exponent prismoid base is a, the edge thickness is $h_A$, the edge height is $H_A$, and the sag width is $r_A$. The thickness change of the power-exponent prismoids in the X and Y directions follows the equations $\begin{cases} H(x) = \varepsilon|x|^m + h_A \\ H(y) = \varepsilon|y|^m + h_A \end{cases}$. Furthermore, the cylindrical hole in the center of the prismoid has a radius of $r_{A1}$ and a depth of $H_{A1}$. The length and width of the thin plate are consistent with those of the lower-end face of the prismoid, and its thickness is $h_B$.

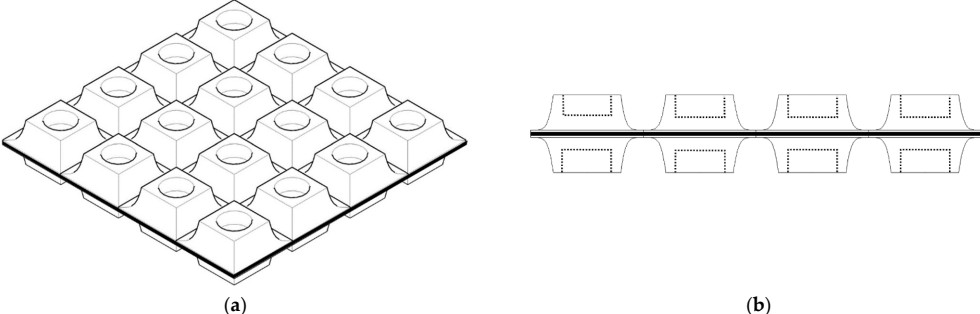

(a) (b)

**Figure 1.** Periodic structure of the photonic crystal: (**a**) Oblique view; (**b**) Front view.

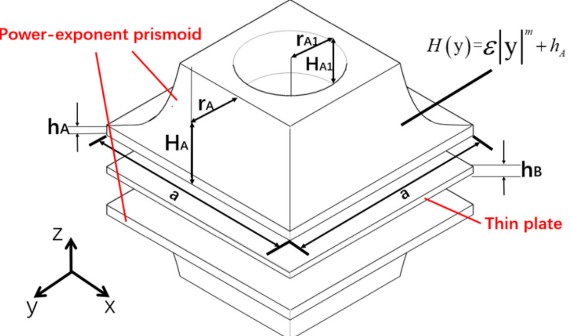

**Figure 2.** Protocell of the photonic crystal.

### 3. Bandgap Characteristics of the Symmetric Power-Exponent Prismatic Phononic Crystal

*3.1. Calculation of the Frequency Dispersion Curves*

In case of geometric linearity, the governing equations for a linear elastic medium can be written in terms of the structural displacement vector $u$ as [29]

$$\nabla \cdot (C : \nabla u) = \rho \frac{\partial^2 u}{\partial t^2} \tag{1}$$

where $\nabla$ is the Hamilton differential sign, and $\rho$ and $C$ are the density tensor and elastic tensor, respectively. For an elastic wave propagating freely in a medium with lattice periodicity, Equation (1) can be re-written as [30]

$$\rho \ddot{u}(r,t) = (\lambda + \mu)\nabla\nabla \cdot u(r,t) + \mu\nabla^2 u(r,t) \tag{2}$$

where $t$ is the time, $r$ is the position vector, and $\mu$ and $\lambda$ are the Lamé constants. According to the Bloch theorem, the displacement field of the periodic structure can be expressed as [30]

$$u(r) = e^{i(k \cdot r)} u_k(r) \tag{3}$$

$$u_k(r) = u_k(r+R) \tag{4}$$

where $k = (k_x, k_y)$ is the Bloch wave vector of the first Brillouin zone, $R$ is the translation vector, and $u_k(r)$ is the vector function with the same periodicity. When the wave equation is solved by finite element software, periodic boundary conditions are applied at the boundaries of the phononic crystals, and the frequency dispersion curves can be calculated for a single cell structure. After meshing the model with finite elements, the eigenvalue equation of a single cell in the discrete form can be expressed as

$$(K - \omega^2 M)u = 0 \tag{5}$$

where $K$ is the stiffness matrix, $M$ is the mass matrix, and $u$ is the displacement matrix of a cell unit. The finite element method is a kind of calculation method that develops rapidly with the development of computer technology. It can be applied to all kinds of complex structures, so it is widely used for engineering calculations. In this paper, the frequency dispersion curves of the phonon crystals are calculated based on finite element software; i.e., COMSOL Multiphysics 5.5 (Svante Littmarck, Stockholm, Sweden). The frequency dispersion curves of the phonon crystals are obtained by changing the Bloch wave vector of the first Brillouin zone, and then solving the eigenvalue problem by finite element software.

Figure 3 displays the irreducible Brillouin region of the phononic crystal. According to the translation periodicity of the protocell, the dispersion curves can be obtained by changing the wave vector $k$ in the irreducible Brillouin region and solving for the eigenvalues via finite element software. The geometrical and material parameters of the proposed phononic crystal are shown in Table 1, in which the thin plate material is steel and the material of the power-exponent prismoid is polycarbonate (PC). Figure 4 provides the calculation results of the dispersion curves of the symmetric power-exponent prismatic phononic crystal.

*3.2. Vibration-Reduction Mechanism Analysis of the Symmetric Power-Exponent Prismatic Photonic Crystal*

Three basic wave modes exist in the plate structures with finite thickness, namely, bending waves, longitudinal waves, and horizontal shear waves. These three modes can be expressed as A mode, S mode, and SH mode, respectively, and are related to their frequency order $n$ ($n \geq 0$), which can be written as An, Sn, and SHn ($n$ = 0, 1, 2 ...), respectively. In order to analyze the wave modes corresponding to the dispersion curves, the displacement components of the eigenmodes at A–G points in the dispersion curves in the $x$, $y$, and $z$ directions are illustrated in Figure 5. Evident from the figure, the vibration amplitudes

of the eigenmodes at the A, B, C, D, and E points in the *z* direction are greater than those in the other two directions (*x* and *y*). The displacement component of the eigenmode of point F in the *y* direction is much larger than that in the other directions, and therefore the wave mode corresponding to the frequency dispersion curve of point F is the horizontal shear wave. Similarly, the displacement component of the eigenmode of point G in the *x* direction is much larger than that in the other directions, so the wave mode corresponding to the frequency dispersion curve of point G is the longitudinal wave [31,32]. Since a thin plate structure produces large vibrations mainly in the vertical direction, the longitudinal wave in the plate that mainly vibrates in the *x* direction, and the horizontal shear wave that mainly vibrates in the *y* direction, are not the focus of this study; thus, the bending wave, which mainly vibrates in the *z* direction, is focused in this study. If other types of wave modes are hidden, it can be found that the proposed symmetric power-exponent prismatic phononic crystal has three complete band gaps of bending waves, and their frequency bands are shown in Table 2.

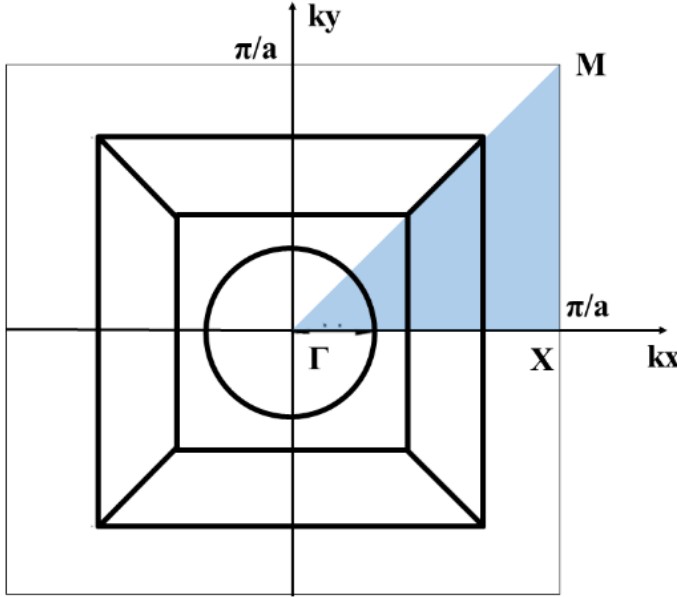

**Figure 3.** Irreducible region of the Brillouin zone of the phononic crystal.

**Table 1.** Geometrical and material parameters of the phononic crystal.

| Parameters | Values |
|---|---|
| Square side length of the prismoid *a* | 50 mm |
| Edge thickness $h_A$ | 0.3 mm |
| Edge height $H_A$ | 15 mm |
| Defect width $r_A$ | 10 mm |
| Power of the power-exponent function *m* | 5 |
| Radius of the cylindrical hole $r_{A1}$ | 11 mm |
| Depth of the cylindrical hole $H_{A1}$ | 10 mm |
| Thin plate thickness $h_B$ | 0.5 mm |
| Young's modulus of the prismoid $E_A$ | 2.048 GPa |
| Density of the prismoid $\rho_A$ | 1200 kg/m³ |
| Poisson's ratio of the prismoid $v_A$ | 0.45 |
| Young's modulus of the thin plate $E_B$ | 210 GPa |
| Density of the thin plate $\rho_B$ | 7800 kg/m³ |
| Poisson's ratio of the thin plate $v_B$ | 0.3 |

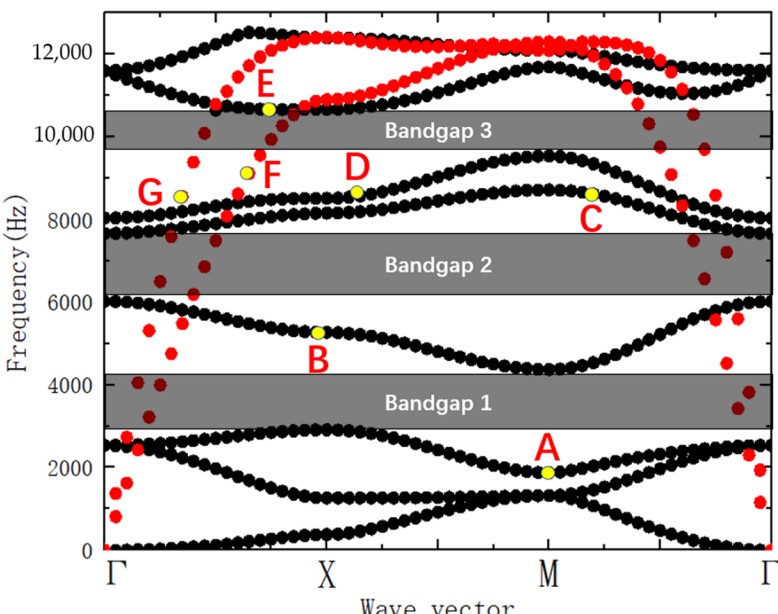

**Figure 4.** Dispersion curves of the photonic crystal.

**Table 2.** Range of complete band gaps of the phononic crystal.

| Band Gap Name | Starting Frequency | Ending Frequency | Bandgap Width |
|---|---|---|---|
| The first band gap | 2903 Hz | 4329 Hz | 1426 Hz |
| The second band gap | 5971 Hz | 7610 Hz | 1639 Hz |
| The third band gap | 9511 Hz | 10,642 Hz | 1131 Hz |

The dispersion curves can be used to predict the band gaps, and it is also necessary to analyze the transmission to verify the band gap characteristics of the phononic crystal, thus fully validating the vibration-reduction characteristics of the phononic crystal plate. As shown in Figure 6, the phononic crystal plate has a length of 0.3 m, a width of 0.25 m, and a thickness of 0.5 mm, and there are 4 × 5 phononic crystal protocells attached to the plate. Moreover, the clamp-supported constraints are applied around the phononic crystal plate, and the geometric and material parameters of the phononic crystal protocell are consistent with those listed in Table 1. Next, a harmonic excitation load $F$ perpendicular to the plate is set at point P1 in the finite element software, and a domain point probe is set at points P1 and P2 in the vibration transmission model, to measure the acceleration amplitudes of the two points. In vibration research, the transmission loss curve is usually defined as

$$H = 20\lg\frac{X_1}{X_0}(\mathrm{dB}) \tag{6}$$

where $X_1$ is the vertical acceleration at point P2, and $X_0$ is the vertical acceleration at point P1.

Based on the vertical acceleration amplitudes of point P1 and P2 calculated by the finite element software, COMSOL Multiphysics 5.5, the transmission loss curve of the vibration transmission model under the excitation frequency of 2000–12,000 Hz can be calculated using Equation (6), as shown in Figure 7. Evidently, compared with the flat plate structure, the phononic crystal plate has a better attenuation effect on the bending waves in the three frequency bands of 2653–4519 Hz, 6097–7888 Hz, and 9085–10,834 Hz. Additionally, the vibration transfer loss of the phononic crystal plate can reach −66 dB, while the energy attenuation bands are basically consistent with the bandgap frequency bands (blue dotted lines in Figure 7).

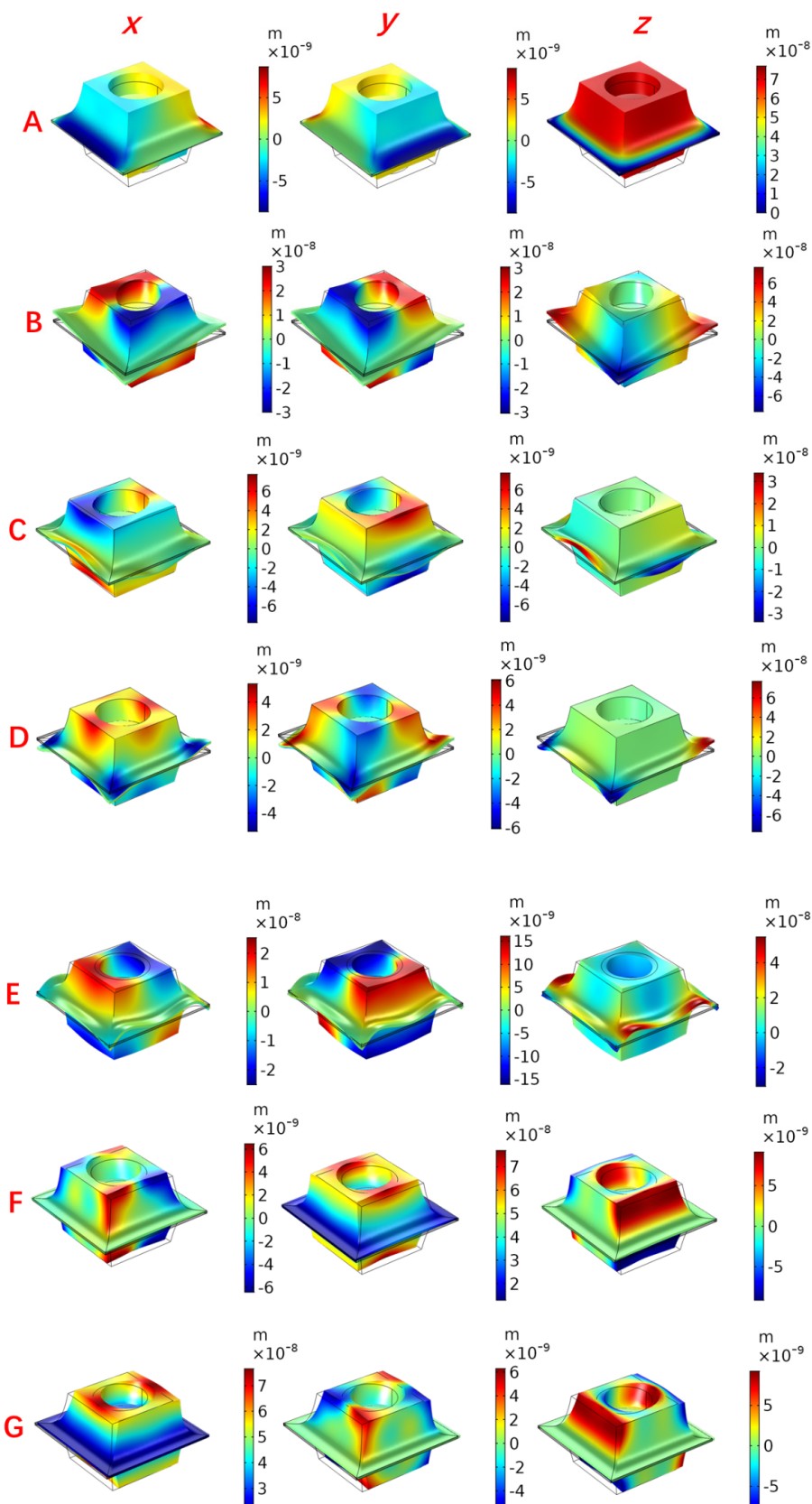

**Figure 5.** Displacement components of the eigenmodes at the (**A**–**G**) points in the dispersion curves in the *x*, *y*, and *z* directions.

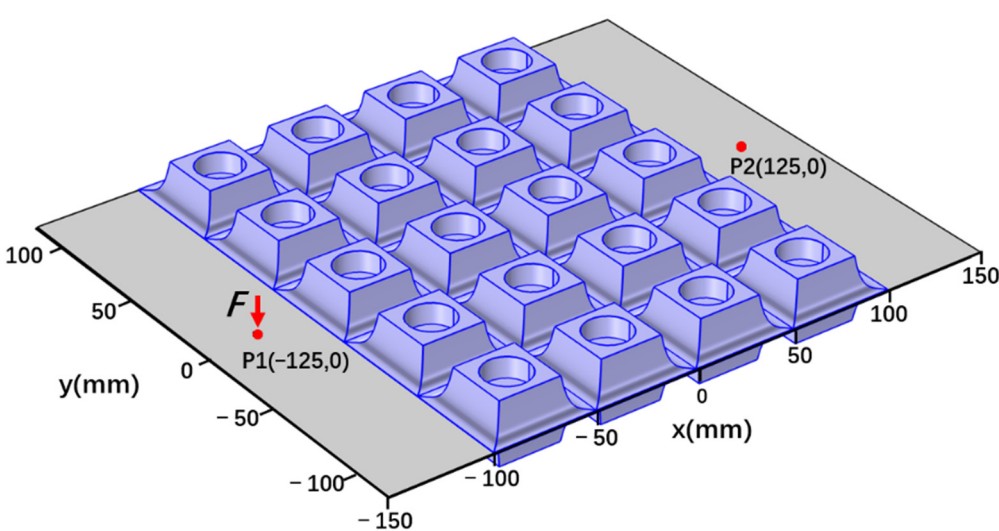

**Figure 6.** Vibration transfer model.

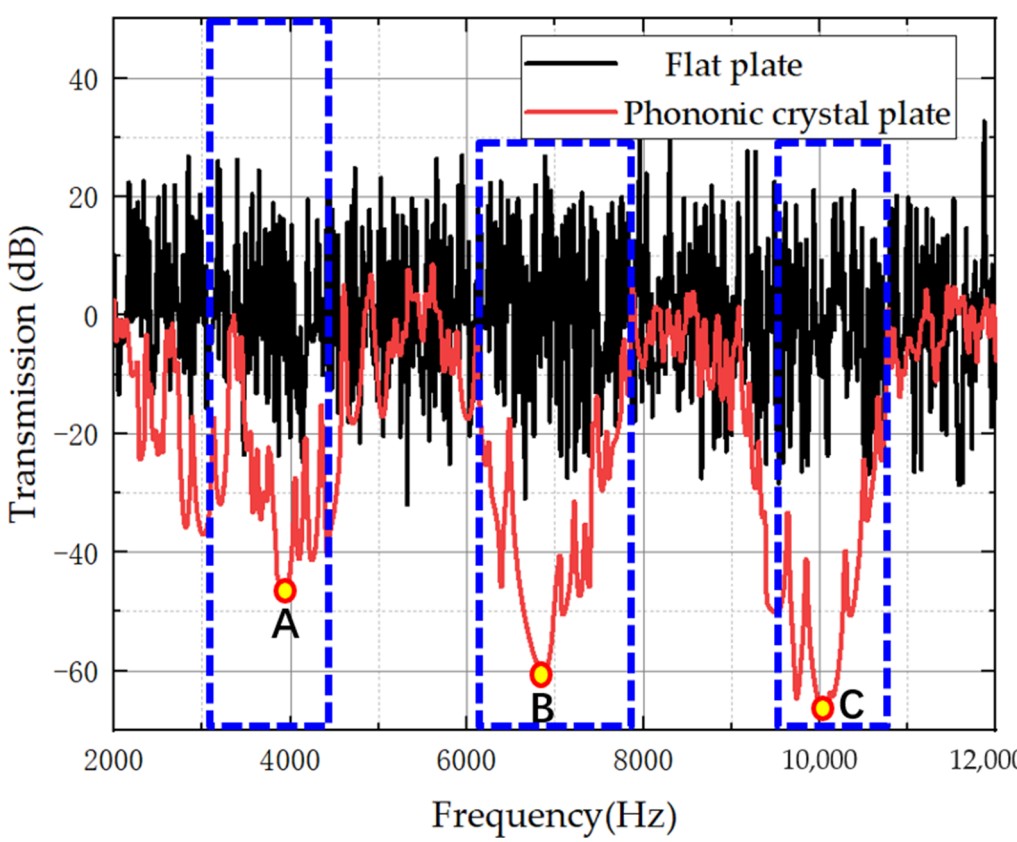

**Figure 7.** Transmission from point P2 to point P1.

To further demonstrate the mechanism of bandgap generation in the symmetric power-exponent prismatic phononic crystal, the displacement fields at the transfer loss peak points, A (3913 Hz), B (6880 Hz), and C (10,066 Hz), in the three attenuation bands were calculated, and the results are reported in Figure 8. Based on the displacement field at point A, it can be observed that the bending vibration of symmetric power-exponent prismatic phononic crystal plate can be effectively suppressed in the frequency bands of the band gaps. From the extracted displacement field of the protocell structure of phononic crystal plate, the phononic crystal plate structure mainly produces the large displacement at the bevel edges of the prismoid.

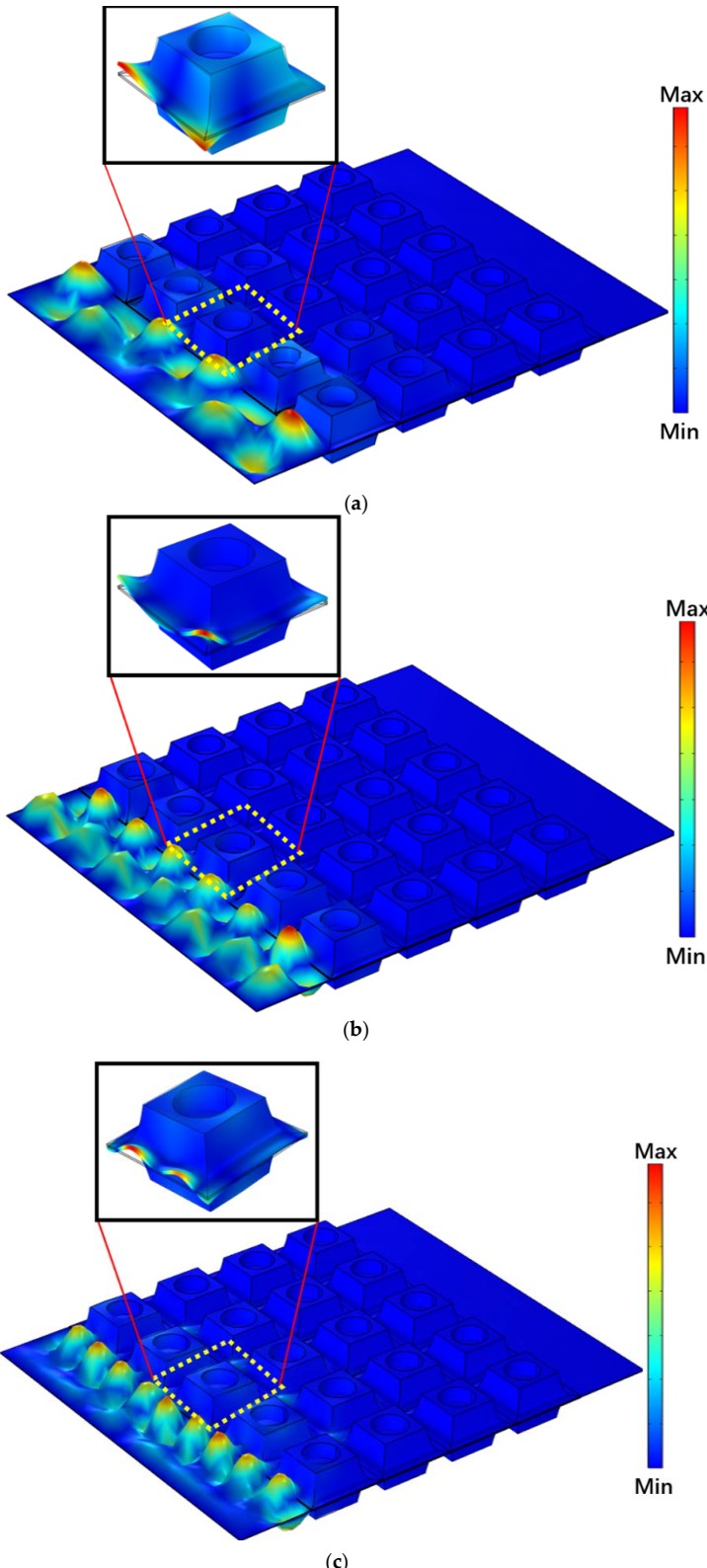

**Figure 8.** Distribution maps of the displacement field: (**a**) response to the excitation at 3913 Hz (Point A in the first band gap); (**b**) response to the excitation at 6880 Hz (Point B in the second band gap); (**c**) response to the excitation at 10,066 Hz (Point C in the third band gap).

In order to better reveal the bandgap mechanism, the natural frequencies of the phonon crystal primitive cell were calculated. Here, the geometric and material parameters

of the phononic crystal are consistent with those listed in Table 1. In the calculation of the natural frequency by the finite element software COMSOL Multiphysics, "solid mechanics" was chosen as the physical field type, "characteristic frequency" was chosen as the research object, and free boundary conditions were chosen as the boundary conditions of the phonon crystal, and its first-order natural frequency is 3923 Hz, corresponding to the frequency band of first band gap. From Figure 9, the vibration mode of the first-order natural frequency of the protocell indicates that the wave energy is mainly concentrated at the bevel edges of the prismoid. Here, the change in prismoid thickness can lead to the decrease of wavelength and the increase in bending wave amplitude, reducing the group velocity and phase velocity of the bending waves, which restricts the energy of the bending waves to the power-exponent bevel faces, i.e., producing the energy-focusing effect [31]. This illustrates that the first band gap is generated by local resonance caused by the energy-focusing effect [22]. Meanwhile, the fifth-order natural frequency of the phononic crystal protocell (6607 Hz) corresponds to the second band gap, and the tenth-order natural frequency of the phononic crystal protocell (9670 Hz) corresponds to the third band gap. The modal shapes of the natural frequencies of protocell provided in Figure 9 all show that the energy-focusing phenomenon occurs on the bevel faces of the power-exponent prismoid. To sum up, three complete band gaps generated by the symmetric power-exponent prismoid are actually generated through local resonance induced by the energy-focusing effect.

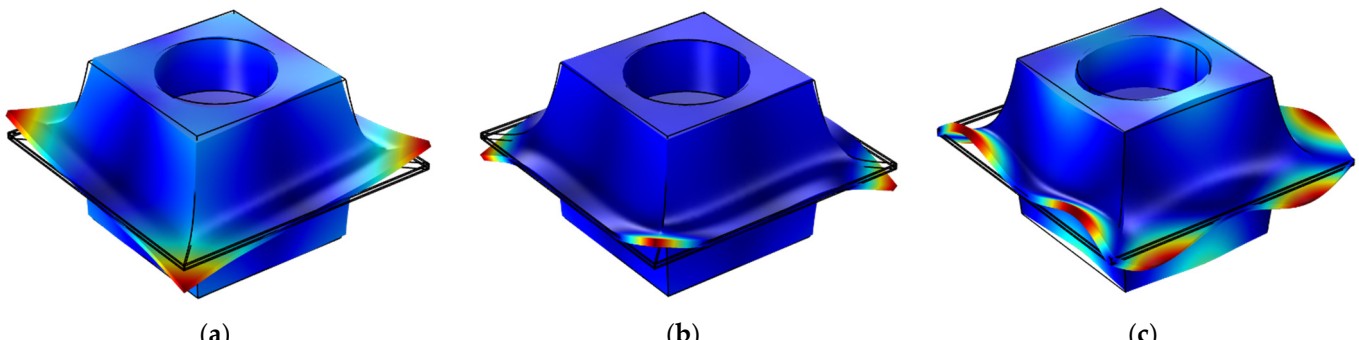

(a)                              (b)                              (c)

**Figure 9.** Modal maps of the natural frequency: (**a**) first natural frequency; (**b**) fifth natural frequency; (**c**) tenth natural frequency.

### 3.3. Experimental Verification

To verify the effectiveness of the band gaps of the symmetric power-exponent prismatic phononic crystal, a test model of the phononic crystal plate was prepared according to the parameters given in Table 1, as shown in Figure 10. The phononic crystal plate was composed of polycarbonate prismoids processed by a Raise3D Pro2 3D printer and one thin steel plate with dimensions of 300 mm × 300 mm × 0.5 mm. In the middle of the thin plate, ten 2 × 5 prismoids were bonded on the upper and lower sides, respectively, using a contact adhesive.

Then, the phononic crystal plate was fixed on a rigid support by bolts to simulate the clamp-supported constraints. Moreover, the vibrator (DH40050) and the lift were placed below the model. The height of the lift was adjusted so that the post rod of the vibration exciter could be placed below point P1 of the phononic crystal plate, to generate the vibration under excitation. A signal generator (FY6900) and a power amplifier (DH5872) were used to set the output excitation force of the vibration exciter. In addition, a miniature acceleration transducer (1A803E) was attached to points P1 and P2 of the phononic crystal plate, to measure the normal vibration acceleration at the two points. Figure 11 shows the overall test system.

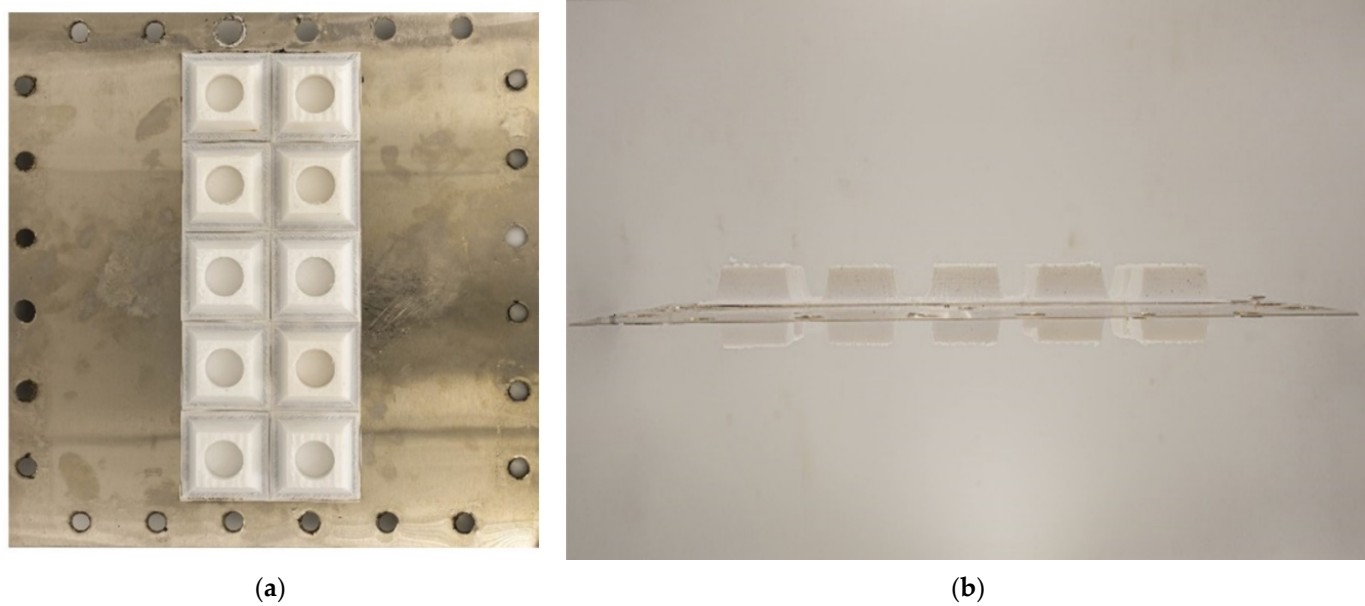

**Figure 10.** Test model of the photonic crystal plate: (**a**) top view; (**b**) front view.

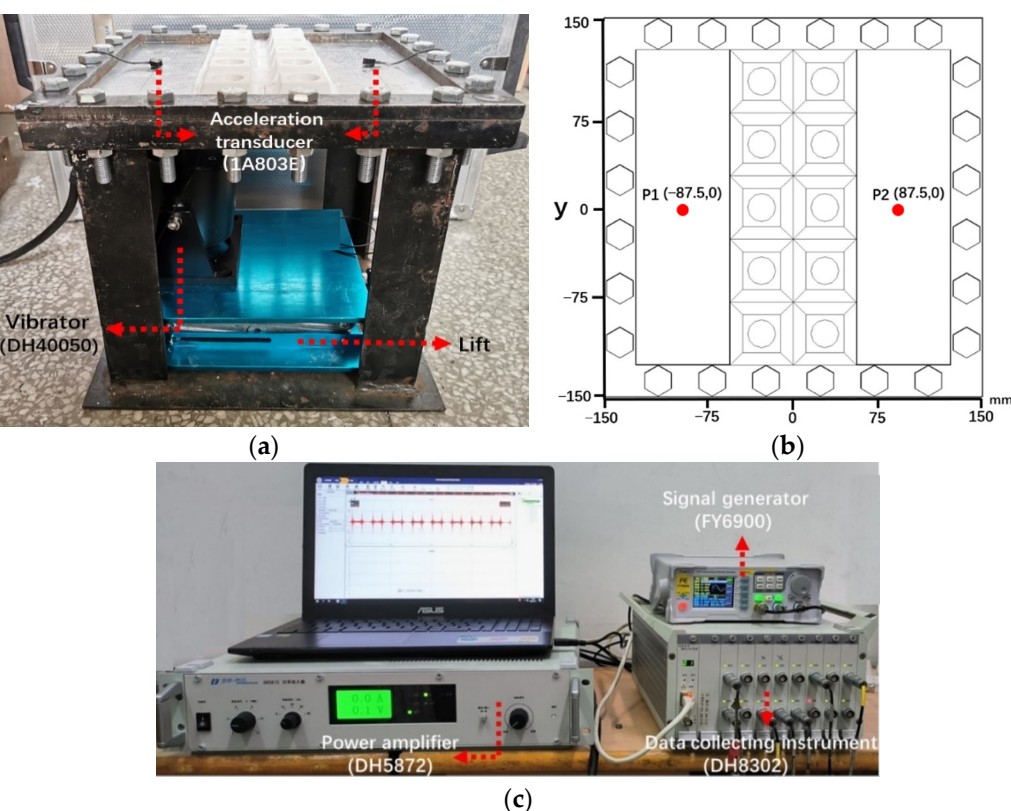

**Figure 11.** Snapshot of the experimental setup: (**a**) overall experimental model; (**b**) local details of the experimental model; (**c**) test device.

The input signals of the vibration exciter in the test are sinusoidal signals with a frequency of 1000~5000 Hz. The fast Fourier transform for the time domain signals of points P1 and P2 was carried out using the software of a date-collecting instrument (DH8302), and the obtained frequency domain signals were then used to calculate the transmission. Figure 12 shows the test results.

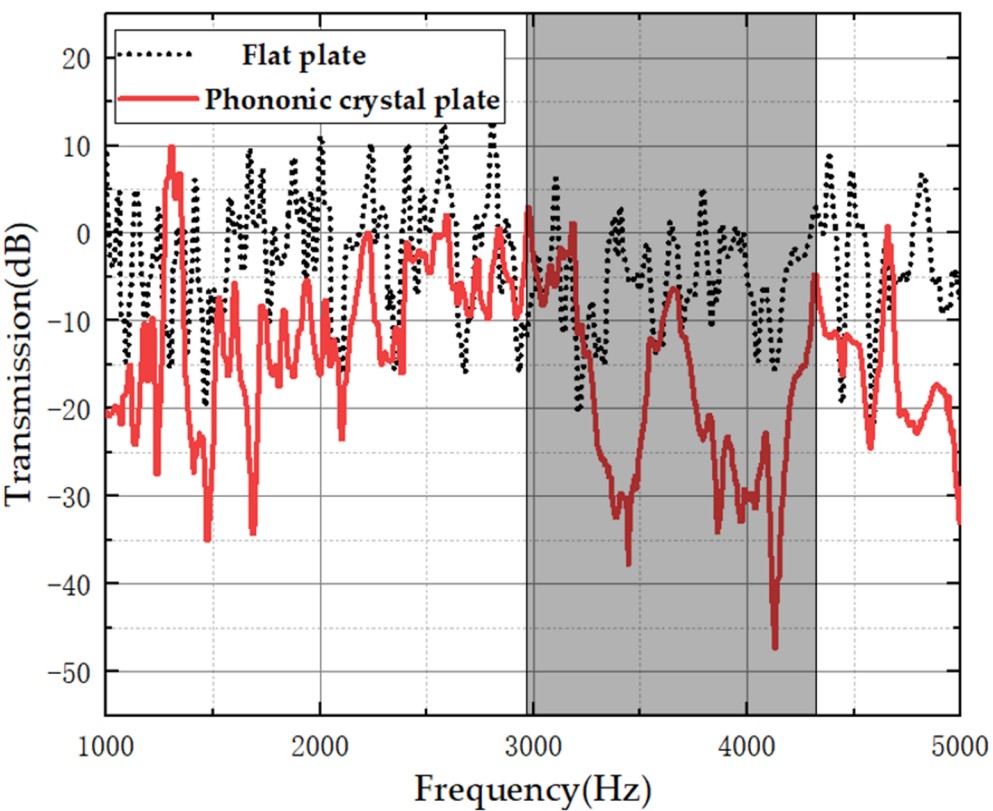

**Figure 12.** Experimental results.

It can be seen from Figure 12 that the test model has an obvious energy attenuation band within the frequency band of 3186–4300 Hz, which effectively suppresses the bending vibration response of the thin plate. Moreover, this energy attenuation band is consistent with the frequency band of the band gaps (the gray zone), thus verifying the effectiveness of suppressing the bending wave vibration through the band gaps of bending waves of the symmetric power-exponent prismatic phononic crystal. In addition, the test sample also has a certain vibration attenuation effect in the frequency band outside the band gaps, since the energy-focusing effect generated by the power exponent profiles of the phononic crystal suppresses the bending vibration of the plate. Furthermore, the experimental results show that the vibration-reduction effect of the phononic crystal plate in the frequency band of the band gaps is better than that in the frequency band without the band gaps.

## 4. Influencing Factors of Band Gaps of the Symmetric Power-Exponent Prismatic Phononic Crystal

To analyze the influence of the structural parameters of phononic crystals on the band gaps, the band gaps of the phononic crystals with different structural parameters are studied in this section.

### 4.1. Influence of Power-Exponent Prismoid Height $H_A$ on the Band Gaps

Figure 13 shows the influence of the height of the power-exponent prismatic phononic crystal on the band gaps. Here, only height $H_A$ of the phononic crystal cell is changed, while the other parameters are consistent with those in Table 1.

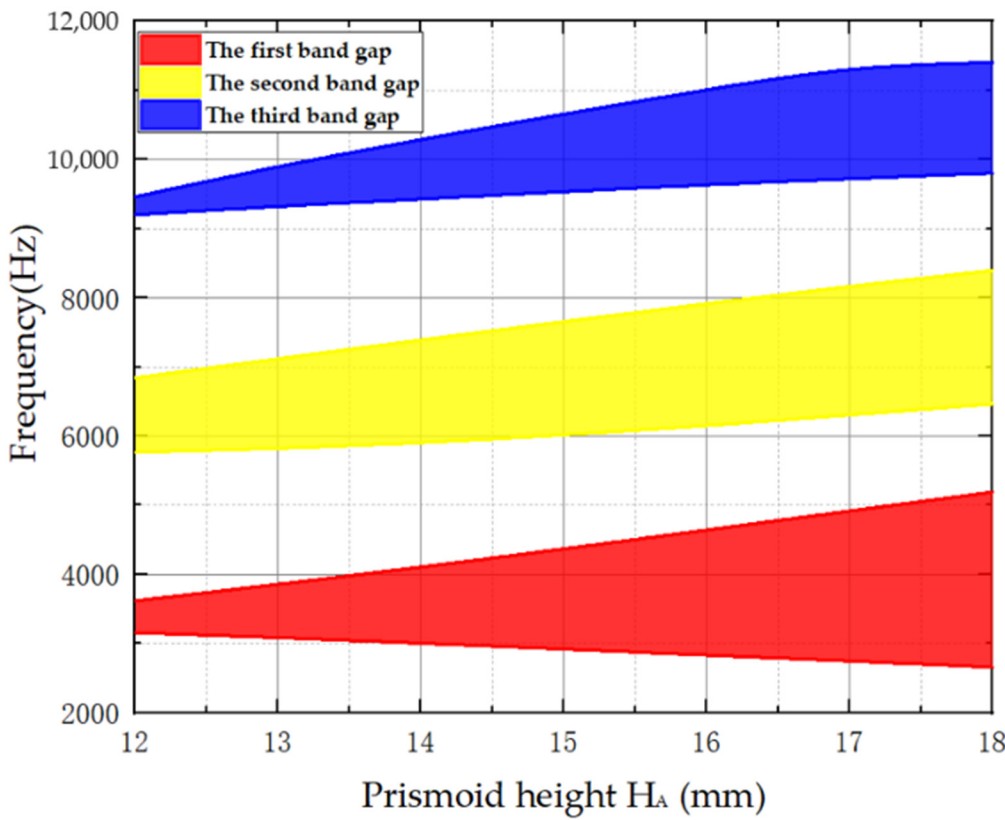

**Figure 13.** Influence of prismoid height on the band gaps.

From Figure 13, with the increase in prismoid height $H_A$, the starting frequency of the first band gap gradually decreases, while that of the other two band gaps increases. Meanwhile, the ending frequencies of the three band gaps all increase with the increase of $H_A$. Notably, the prismoid height $H_A$ has the most significant influence on the band gap range of first band gap, and the bandwidths of all three bandgaps become wider with the increase in $H_A$.

### 4.2. Influence of Power Number m of the Power-Exponent Prismoid Edge on Band Gaps

Figure 14 demonstrates the effect of power number of power-exponent prismatic phononic crystal on the band gaps. Here, only the power number $m$ of the phononic crystal cell is changed, and the other parameters are consistent with those in Table 1.

Evident from Figure 14, with the increasing power number, the frequency bands of the three bandgaps begin to narrow down. This is because the increase in power number results in a reduction in structural stiffness, thus reducing the resonant frequency. Hence, band gaps with lower frequency and wider bandwidth can be obtained by increasing the power number of the symmetric power-exponent prismatic phononic crystal. It is worth mentioning that the band gap starts to appear only when the power number $m$ is greater than 2, because the energy-focusing effect is triggered when this condition is satisfied [33].

### 4.3. Influence of Prismoid Edge Thickness $h_A$ on the Band Gaps

Fundamentally, the band gaps of the power-exponent prismatic phononic crystal are very sensitive to its edge thickness. Likewise, Figure 15 shows the influence of the prismoid edge thickness of the power-exponent prismatic phononic crystal on the band gaps.

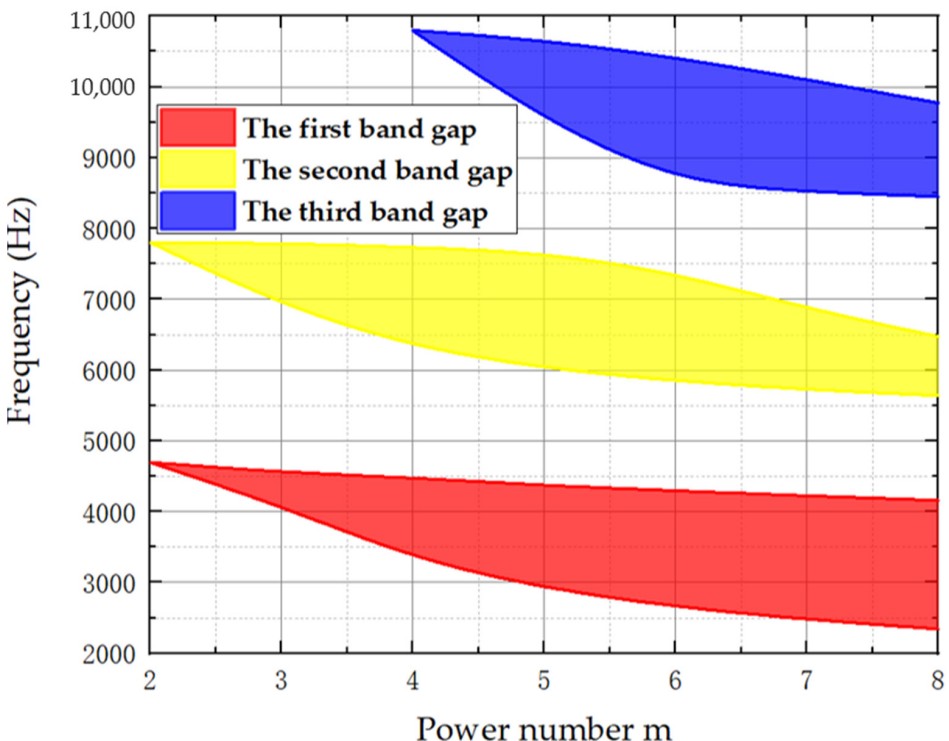

**Figure 14.** Influence of power number of the power-exponent prismoid edge on the band gaps.

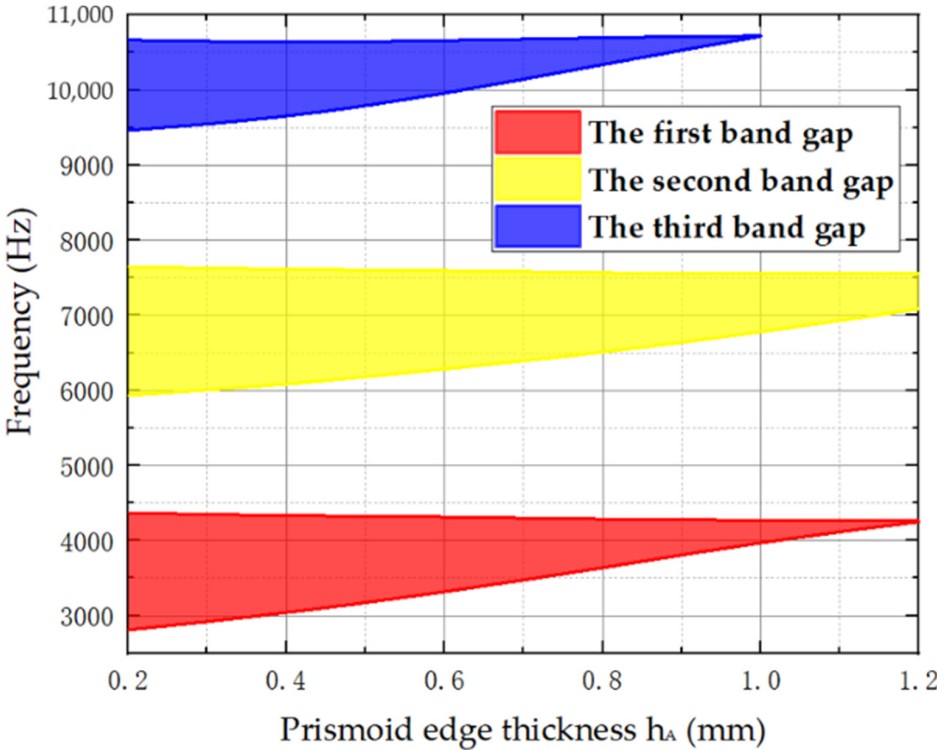

**Figure 15.** Influence of prismoid edge thickness on the band gaps.

From Figure 15, as the edge thickness $h_A$ increases, the frequency bands of all the band gaps gradually narrow down, and the starting frequency of the band gaps begins to increase. When $h_A = 1.2$ mm, the first band gap disappears. This is because the energy-focusing effect weakens as the edge thickness increases, which in turn leads to the weakening of

local resonance effect [31]. Hence, band gaps with a lower starting frequency and wider bandwidth can be obtained by reducing the prismoid edge thickness.

## 5. Conclusions

In summary, a symmetric power-exponent prismatic phononic crystal configuration is proposed, and the characteristics of the band gaps of the bending waves were studied. The main conclusions drawn from this work are as follows:

(1) The proposed symmetric power-exponent prismatic phononic crystal structure has three wide-band bending-wave band gaps. The calculation results demonstrate that the 1st, 5th, and 10th natural frequencies of the phononic crystal protocell correspond to the frequency bands of the 1st, 2nd, and 3rd band gaps, respectively. In addition, the modal shapes show that the energy-focusing phenomenon occurs at the bevel faces of the prismoid. Therefore, the band gaps of the phononic crystals are generated by the local resonance caused by the energy-focusing effect. The bandgap characteristics of the symmetric phononic crystal are verified by numerical simulation and experimental test, and it can be concluded that the bending vibration of a thin plate can be effectively suppressed via the energy-focusing effect and the bandgap characteristics of the phononic crystal.

(2) With an increasing height, $H_A$, of the power-exponent prismoid, the bandwidths of all three band gaps become wider; the starting frequency of the first band gap begins to decrease, while that of the other two band gaps gradually increases. Moreover, the increase in power number $m$ of the prismoid can lead to a decrease in the starting frequencies of the band gaps. In addition, the increase of prismoid edge thickness can weaken the energy-focusing effect, and thus the band gaps gradually narrow down. On the premise of ensuring sufficient structural strength of the plate, the edge thickness, $h_A$, can be reduced to obtain band gaps with a lower starting frequency and wider bandwidth.

**Author Contributions:** Conceptualization, X.J. and Z.Z.; methodology, X.J. and Z.Z.; validation, X.J. and Z.Z.; writing—original draft preparation, X.J.; supervision, Z.Z.; funding acquisition, Z.Z. All authors have read and agreed to the published version of the manuscript.

**Funding:** The research was funded by National Natural Science Foundation of China, grant number 51879270.

**Institutional Review Board Statement:** Not applicable.

**Informed Consent Statement:** Not applicable.

**Data Availability Statement:** The data that support the findings of this study are available from the corresponding author upon reasonable request.

**Conflicts of Interest:** The authors declare no conflict of interest.

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
