# Peer review of "Study on the Bandgap Characteristics and Vibration-Reduction Mechanism of Symmetric Power-Exponent Prismatic Phononic Crystal Plates"

_crystals, doi:10.3390/cryst12081125_

Round 1
Reviewer 1 Report
The authors proposed a phononic crystal for bending waves on a thin plate. The phononic crystal consists of periodic power-exponent prismatic on both side of the thin plate. They calculated the band structure of a phononic crystal, numerically calculated the transmission, and verified the results by experiment.
There are several points that may need to be revised.
1. In line 121, the h in the equation should be hA or hB.
2. Does the finite element solver solve eq(1) in the band gap calculation? I suggest the authors provide some more detail about band structure (Figure 4) calculation.
3. Although the modes E and F are not the major concerns of this manuscript, I still suggest the author add some descriptions in line 168 according to ref 29, rather than simply calling the modes “other wave modes”.
4. Is the results of Figure 7 obtained by simulation? The numerical description seems missing.
5. In the paragraph starting from line 212, the authors mentioned the “natural frequency of the phononic crystal protocell is also calculated”. Please explain what is natural frequency and descript the calculation process.
6. In the paragraph starting from line 212, the authors mentioned that the first band gap results from the first-order natural frequency, and the second band gap results from the fifth-order natural frequency. However, the 2nd, 3rd, and 4th natural frequencies are not mentioned. Why didn’t 2nd, 3rd, and 4th natural frequencies generate band gaps?
7. I suggest the authors descript more clearly the “natural frequency calculation”. Is the periodic boundary condition applied in the simulation?
8. The authors mentioned that the proposed phononic crystal is based on the ABH configuration. I suggest the authors should mention the role of ABH somewhere in this manuscript.
Reviewer 2 Report
The authors study numerically and experimentally bending vibrations in thin plates with periodic surface 2D corrugation. The periodic structure is formed by irregularities having a symmetric power-exponent prismatic shape. Such a structure is proposed with the aim of reducing bending vibrations of plates. In particular, it has been shown that the proposed phononic crystal structure has three complete band gaps. The band gaps of bending waves were analyzed using numerical simulations and experimental methods.
I think the paper can be published after revision.
11. Which numerical method is used?
22. Please write whether a homemade program or a commercial package is used.
33. Please give the reference to a publication where Eq. (1) has been derived for the case of plates with varying thickness.
44. Figures 4 and 5 display results concerning not only bending vibrations but also in-plane ones, so it would be logical to give not only the equation for bending vibration but also the equations describing in-plane vibrations.
Author Response
We are truly grateful for your kind comments and thoughtful suggestions. We had upload the reponse in the attachment.Please see the attachment.
